# Ventromedial prefrontal cortex stimulation enhances memory and hippocampal neurogenesis in the middle-aged rats

**Albert Liu[1], Neeraj Jain[1], Ajai Vyas[1], Lee Wei Lim[1,2]\***

[1]School of Biological Sciences, Nanyang Technological University, Singapore, Singapore; [2]Department of Biological Sciences, Sunway University, Bandar Sunway, Malaysia

**Abstract** Memory dysfunction is a key symptom of age-related dementia. Although recent studies have suggested positive effects of electrical stimulation for memory enhancement, its potential targets remain largely unknown. In this study, we hypothesized that spatially targeted deep brain stimulation of ventromedial prefrontal cortex enhanced memory functions in a middle-aged rat model. Our results show that acute stimulation enhanced the short-, but not the long-term memory in the novel-object recognition task. Interestingly, after chronic high-frequency stimulation, both the short- and long-term memories were robustly improved in the novel-object recognition test and Morris water-maze spatial task compared to sham. Our results also demonstrated that chronic ventromedial prefrontal cortex high-frequency stimulation upregulated neurogenesis-associated genes along with enhanced hippocampal cell proliferation. Importantly, these memory behaviors were strongly correlated with the hippocampal neurogenesis. Overall, these findings suggest that chronic ventromedial prefrontal cortex high-frequency stimulation may serve as a novel effective therapeutic target for dementia-related disorders.

**\*For correspondence:** drlimleewei@gmail.com

## Introduction

Memory loss is the key symptom of dementia-related disorders along with impaired cognitive functioning such as language or reasoning. It is usually caused by Alzheimer's disease and other age-related dementia. Its prevalence doubles from a low rate in 60–64 age group to 40–50% of those older than 85 (*Lobo et al., 2000*). Dementia is a progressive disease, which has a detrimental impact on the quality of life for patients. To date, pharmacological treatments for dementia have limited effects and there are no known treatments that cure or delay the progression of this memory impairment (*Doody et al., 2014*; *Salloway et al., 2014*). Therefore, a novel non-pharmacological approach such as deep brain stimulation (DBS) is currently considered as an alternative treatment to reduce the symptomatic and progression of this memory deterioration (*Hescham et al., 2013a*).

DBS, a technique of minimally invasive surgical implantation of electrodes with delivering of electrical impulses into the brain, has been demonstrated to control a wide range of neurological disorders and neuropsychiatric diseases (*Sesia et al., 2009*; *Temel et al., 2012b*; *Temel and Lim, 2013*). In line with these developments, evidence from recent studies suggests that DBS might enhance memory functions when particular brain areas are stimulated (*Hamani et al., 2008*; *Laxton et al., 2010*; *Suthana et al., 2012*; *Hescham et al., 2013b*). Of particular interest, DBS of the subgenual anterior cingulate cortex or the ventromedial prefrontal cortex (vmPFC) induced striking antidepressant activity in both patients and animal studies (*Mayberg et al., 2005*; *Lozano et al., 2008*;

**eLife digest** Memory loss in older people is a serious and widespread problem that affects up to 50% of those over the age of 85. It is a key symptom of dementia, but despite the growing impact of this disease on society, there are no treatments currently available that can effectively stop or delay the progression of the symptoms.

One therapy that may reduce memory loss is called deep brain stimulation. Electrodes are implanted into the brain and used to stimulate brain cells in particular areas of the brain to alter mental and emotional processes. Deep brain stimulation is already used to treat Parkinson's disease, depression and other conditions that affect how the brain works.

Liu et al. studied the effect of deep brain stimulation on memory in rats. The experiments show that middle-aged rats performed less well in short- and long-term memory tests than young rats. Next, Liu et al. investigated whether deep brain stimulation could improve memory in the middle-aged rats. The electrodes were positioned to stimulate a region near the front of the brain called the 'ventromedial prefrontal cortex'; this region is important for the formation and recall of memories. Liu et al. then gave the rats a series of memory tasks that tested their recall after 90 minutes (to test their short-term memory), and after 24 hours (to test long-term memory).

The experiments reveal that a brief stimulation of brain cells in this region of the brain improved the rats' short-term memory, but not their long-term memory. However, more sustained stimulation of this region of the brain improved both the short-term and long-term memory of the rats. Furthermore, deep brain stimulation led to the formation of new brain cells in another region of the brain called the hippocampus, which is also involved in memory. The hippocampus had not been in direct contact with the electrodes so the increase in brain cells was due to its connections with the stimulated ventromedial prefrontal cortex.

Liu et al.'s findings suggest that deep brain stimulation of the ventromedial prefrontal cortex has the potential to be developed into a therapy to treat dementia and other conditions that lead to memory loss in humans.

---

*Hamani et al., 2010b*; *Kennedy et al., 2011*; *Temel and Lim, 2013*; *Lim et al., 2015b*). Despite encouraging results, no studies have shown the putative role of vmPFC DBS in learning and memory performance. In the realm of cognitive function, there is empirical evidence indicating that vmPFC plays an important role in the formation, consolidation, and retrieval of memory, as well as reward and decision making (*Maviel et al., 2004*; *Euston et al., 2012*). Based on the human imaging and rodent studies, the vmPFC was significantly activated during the recall of remote memory (*Bontempi et al., 1999*; *Takashima et al., 2006a*, *2006b*; *Gais et al., 2007*), while its inactivation caused memory impairment when tested in the radial arm-maze (*Maviel et al., 2004*), the Morris water-maze (MWM) (*Teixeira et al., 2006*), and the contextual fear conditioning (*Frankland et al., 2004*). In line with these studies, malfunctioning has also been reported in the hippocampus and the vmPFC (which received robust projections from the hippocampal formation) in early stages of Alzheimer's disease, frontotemporal dementia, and healthy aging-related memory impairments (*Salat et al., 2001*; *Lindberg et al., 2012*).

Given the potential mechanisms involved by DBS including the increase of hippocampal brain-derived neurotrophic factor (BDNF) levels (*Hamani et al., 2012*; *Ying et al., 2012*) and neurogenesis-related functions (*Toda et al., 2008*; *Kadar et al., 2011*; *Stone et al., 2011*), we tested the hypothesis that vmPFC DBS-enhanced memory function by modulating the hippocampal neurogenic activity in the middle-aged rat model with aging-related memory impairment. The use of this animal model was supported by previous data that showed aged-related deficits in both the memory and the hippocampal functioning (*Rex et al., 2005*; *Kaczorowski and Disterhoft, 2009*). In acute DBS, animals were tested with either high- or low-frequency stimulation (HFS or LFS) at various amplitudes using the conventional novel-object recognition (NOR) test. Subsequently, another set of animals was used to assess the chronic stimulation effects on memory enhancement using the NOR and the MWM tests. For investigation of the underlying mechanism, we analyzed the effects of chronic stimulation on the molecular and cellular levels of hippocampal neurogenesis-related functions.

## Results

### Memory deficits in middle-aged rats

Progressive age-related memory decline has been previously described for human (*Davis et al., 2003*) and animal studies (*Sloane et al., 1997*; *Ward et al., 1999*; *Kaczorowski and Disterhoft, 2009*). We compared the short- and long-term memory functions in young (n = 20) and middle-aged (n = 15) rats using the NOR test (*Figure 1A*). Three-way ANOVA (group age × retention interval × object) with repeated-measures showed significant effects for object ($F_{(1,82)} = 18.043$, $p < 0.001$), retention interval ($F_{(2,82)} = 13.956$, $p < 0.001$), and the interaction group age × retention interval × object ($F_{(1,82)} = 4.160$, $p = 0.019$) (*Figure 1C,D*). No differences were observed for group ($F_{(1,82)} = 0.009$, $p = $ n.s.). With regard to the duration of object exploration, there was no significant difference between the young and middle-aged rats in the acquisition phase ($t_{(28)} = -0.742$, $p = $ n.s.), see *Supplementary file 1A*. However, a decrease in the duration of novel object exploration was observed for the middle-aged group when compared to the young in the long-term ($t_{(26)} = 4.129$, $p < 0.001$), but not the short-term ($t_{(29)} = 0.014$, $p = $ n.s.) memory. Interestingly, the young animals spent relatively more time with the novel object as compared to the familiar object in both the short- ($t_{(18)} = -5.23$, $p < 0.001$) and long-term ($t_{(14)} = -8.722$, $p < 0.001$) phase (*Figure 1C,D*). In the middle-aged rats, no significant effect was found for discrimination between the novel and familiar objects in the short- and long-term memory

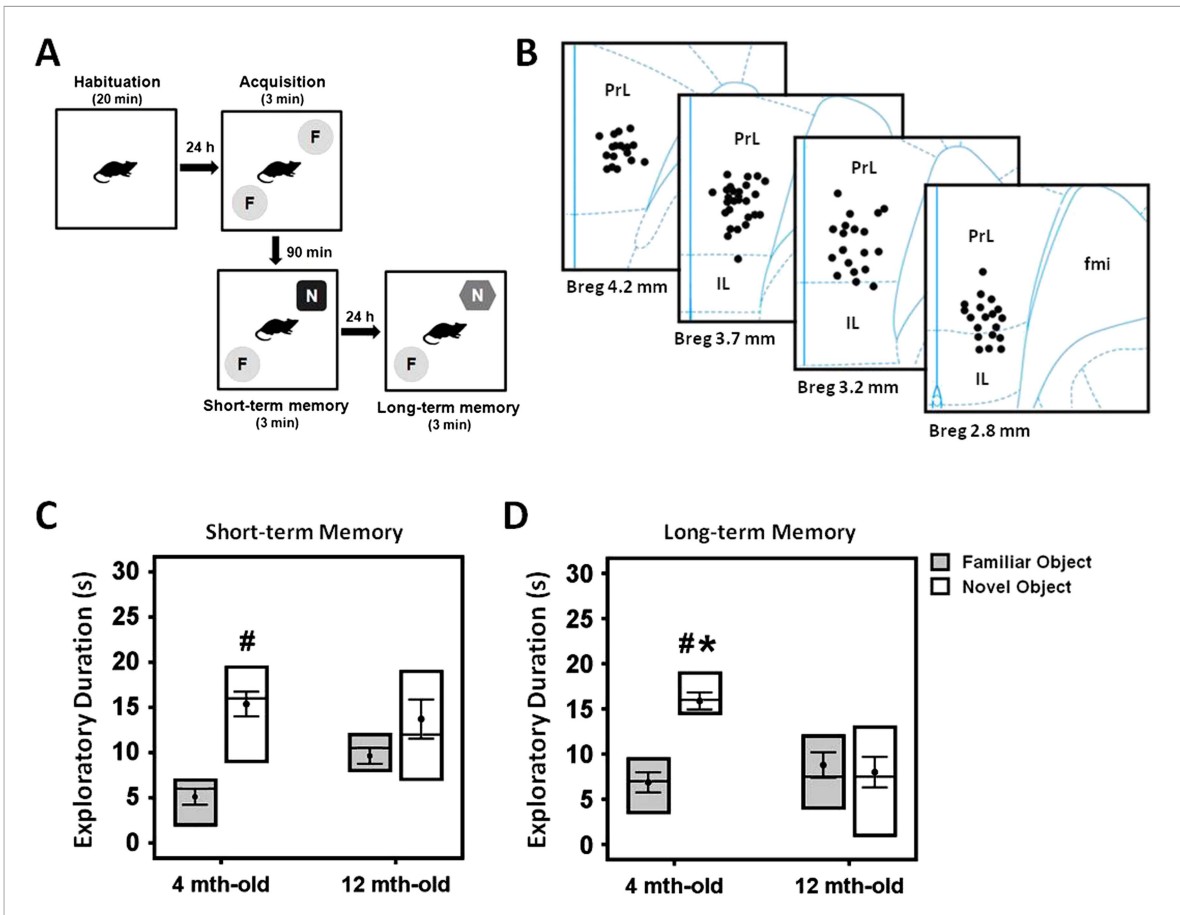

**Figure 1**. Experimental protocol of the novel-object recognition test (**A**), and representative illustration of the stimulating electrode localization in the vmPFC (**B**). The box plots show the comparisons between young (4 month old) and middle-aged (12 month old) animals on the short- and long-term memory retention interval in the novel-object recognition task (**C**, **D**). Note: there was a decrease of time spent in the novel object exploration in the middle-aged animals as compared to the young rats, suggesting a possible manifestation of memory deficit in this animal model. Indication: *, significant difference from the middle-aged rats; #, significant difference from the familiar object of respective age animals, ($p < 0.05$).

retention interval (all $t_{(11)} > -2.058$, p = n.s.), indicating a possible manifestation of memory deficit in the middle-aged animals.

## Acute stimulation enhanced the short-, but not the long-term memory

To test the hypothesis that electrical stimulation enhances memory functions, the middle-aged animals were stereotactically implanted with electrodes in the vmPFC region. All localization of electrode tips were verified within the vmPFC target as illustrated in a representative *Figure 1B*. Animals with electrode misplacement or detachment in the acute stimulation (HFS: 50 µA, n = 2; 100 µA, n = 2; 400 µA, n = 3; and LFS: 50 µA, n = 1; 400 µA, n = 2; and Sham, n = 4) and chronic stimulation (vmPFC HFS, n = 3; Sham, n = 3) experiments were excluded from data analysis. Overall, the final number of rats per group was as follows for the acute stimulation with either HFS (50 µA, n = 8; 100 µA, n = 8; 200 µA, n = 10; and 400 µA, n = 7) or LFS (50 µA, n = 11; 100 µA, n = 12; 200 µA, n = 12; and 400 µA, n = 8) in comparison with the sham animals (n = 12). In chronic stimulation experiment, the final number of rats per group was as follows for the vmPFC HFS (n = 12) and the sham (n = 9) animals.

For determination of acute stimulation efficacy in memory functions, animals were tested with either HFS or LFS at amplitudes varying across 50, 100, 200, and 400 µA. Both the short- and long-term memory functions were assessed using the NOR test. In the HFS animals, repeated-measures three-way ANOVA (group × retention interval × object) showed main effects for group ($F_{(4,116)} = 5.873$, p < 0.001), retention interval ($F_{(2,116)} = 16.670$, p < 0.001), object ($F_{(1,116)} = 9.552$, p = 0.003), and the interaction group × retention interval × object ($F_{(8,116)} = 2.342$, p = 0.023). Similarly, in the LFS animals, there were significant effects for object ($F_{(1,145)} = 63.815$, p < 0.001), retention interval ($F_{(2,145)} = 16.418$, p < 0.001), group ($F_{(4,145)} = 2.544$, p = 0.042), and interaction group × retention interval × object ($F_{(8,145)} = 3.112$, p = 0.003). In the acquisition phase, no differences were found in the duration of object exploration for both the HFS ($F_{(4,40)} = 1.509$, p = n.s.) and the LFS ($F_{(4,50)} = 0.478$, p = n.s.) groups, see *Supplementary file 1B,C*. In acute HFS, stimulation at 200 µA significantly increased the duration of novel object exploration in the short- ($F_{(4,39)} = 7.995$, p < 0.001), but not the long-term ($F_{(4,39)} = 1.553$, p = n.s.) memory when compared to the sham (*Figure 2A,B*). The novel object exploration was also higher compared to the familiar object ($t_{(9)} = -14.636$, p < 0.001). In acute LFS, stimulation at 50, 200, and 400 µA induced a longer duration of novel object exploration in the short-term memory ($F_{(4,49)} = 4,432$, p = 0.004) when compared to the sham (*Figure 2C,D*). As for the long-term memory, Bonferroni post-hoc test revealed no significant difference for the novel object exploration between groups. Interestingly, for comparisons of discrimination between novel and familiar objects, LFS at 50, 100, 200, and 400 µA (all $t_{(6-11)} > -6.250$, p < 0.001) increased the duration of novel object exploration in the short-term memory. In terms of the long-term memory, an increase for novel object exploration was found with LFS at 50, 100, and 200 µA (all $t_{(9-11)} > -3.440$, p < 0.029), but not at 400 µA (all $t_{(6)} = 0.969$, p = n.s.) when compared to the familiar object.

## Chronic stimulation facilitated long-lasting effects on learning and memory

### Novel-object recognition test

For chronic stimulation experiment, we used stimulation parameter of HFS (100 Hz) at amplitude 200 µA, which was based on the data derived from the present (*Figure 2*) and previous studies showing memory enhancement and antidepressant effects (*Hamani et al., 2010a*; *Lim et al., 2015b*). In the NOR test, the effects of chronic stimulation were measured in two different conditions with respect to the animals tested with either no-HFS or HFS prior to the behavioral testing. Using four-way ANOVA (group × retention interval × condition × objects) with repeated-measures, there were significant differences for group ($F_{(1,74)} = 24.140$, p < 0.001), retention interval ($F_{(1,74)} = 9.371$, p = 0.003), objects ($F_{(1,74)} = 18.304$, p < 0.001), and the interaction group × retention interval × condition × objects ($F_{(5,74)} = 3.818$, p = 0.004). No effect was observed for condition ($F_{(1,74)} = 0.155$, p = n.s.), indicating that the order of testing did not affect the behaviors in both conditions. In the acquisition phase, we found no differences in the exploratory duration for animals that tested in conditions with either no-HFS ($t_{(15)} = 1.531$, p = n.s.) or HFS ($t_{(14)} = 1.272$, p = n.s.) prior to the NOR task, see *Supplementary file 1D,E*. After chronic stimulation, animals that tested with no-HFS prior to the behavioral testing, spent significantly longer duration exploring novel object in the short- ($t_{(16)} = 2.981$, p = 0.009), but not the long-term ($t_{(15)} = 1.656$, p = n.s.) retention intervals (*Figure 3A,B*)

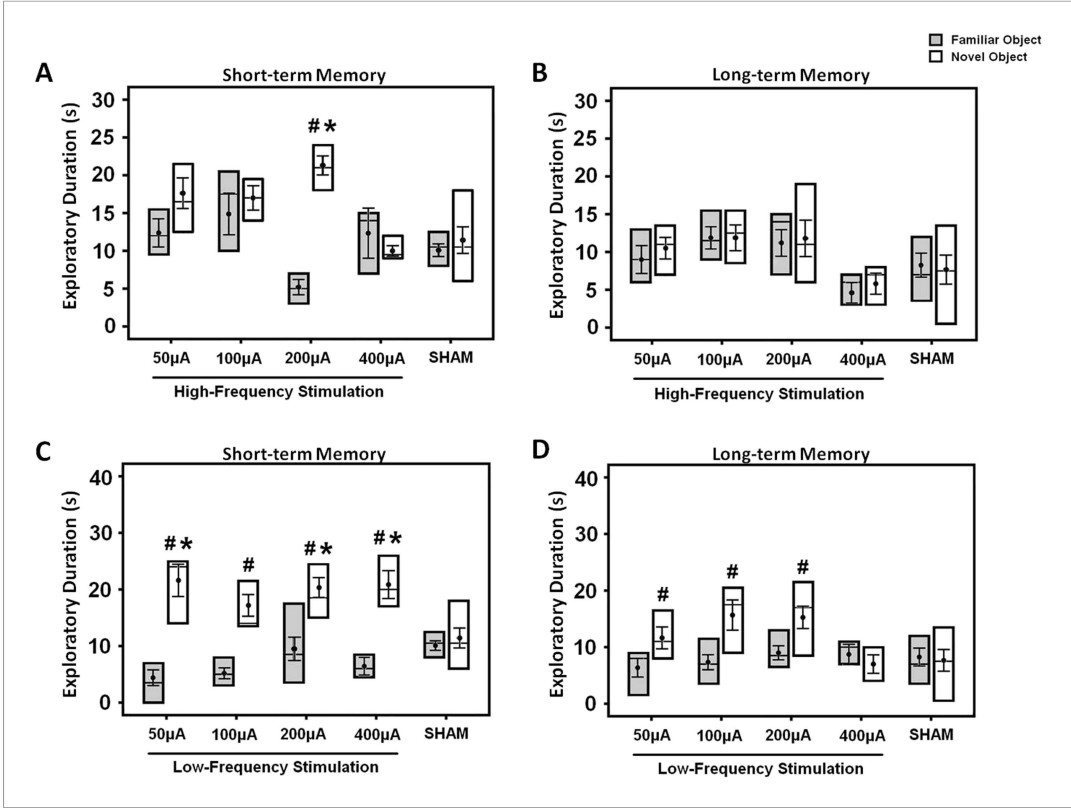

**Figure 2**. The box plots show the effects of either high- (**A**, **B**) or low-frequency (**C**, **D**) stimulation at amplitudes varying across 50, 100, 200, and 400 µA in the middle-aged animals. Both the short- and long-term memory functions were tested using the novel-object recognition test. Note: HFS (100 Hz) at 200 µA and LFS (10 Hz) at 50, 200, 400 µA significantly increased the novel object exploration as compared to the sham animals, respectively. Indication: *, significant difference from the sham rats; #, significant difference from the familiar object of respective stimulation amplitude, ($p < 0.05$).

when compared to the sham group. For comparisons between the novel and familiar objects, chronic stimulation induced a significantly higher exploration time for the novel object in the short- ($t_{(9)} = -6.710$, $p < 0.001$), but not the long-term memory ($t_{(8)} = -1.891$, $p = 0.095$). Interestingly, animals that were treated with HFS prior to the behavioral testing, showed remarkable increase in the novel object exploration for both the short- ($t_{(15)} = 2.686$, $p = 0.017$) and long-term ($t_{(14)} = 3.783$, $p = 0.002$) intervals as compared to the sham (*Figure 3C,D*). The duration for novel object exploration compared to the familiar object was also significantly increased in both the short- and long-term memory (all $t_{(9)} < -6.190$, $p < 0.01$) in animals with vmPFC HFS prior to the NOR testing. No significant difference was found in the exploration time for familiar object in both the short- and long-term memory (all $t_{(>14)} < 0.865$, $p = $ n.s.) between the vmPFC HFS and sham animals.

For the results of the total exploratory duration of the identical objects during the acquisition phase for animal experiments of comparison between the young and middle-aged rats, acute stimulation, and chronic studies, see *Supplementary file 2A,B,C*.

## Morris water-maze test

Repeated-measures ANOVA showed significant main effects for group ($F_{(1,15)} = 12.454$, $p = 0.003$), day ($F_{(3,45)} = 149.689$, $p < 0.001$), and the interaction group × day ($F_{(3,45)} = 3.509$, $p = 0.023$), indicating variations in the ability of animals to locate the submerged platform. When analyzing the training phase from days 1 to 4, animals treated with vmPFC HFS exhibited a shorter latency to reach the hidden platform on day 1 ($t_{(17)} = -2.348$, $p = 0.031$) and day 2 ($t_{(17)} = -2.810$, $p = 0.012$) as compared to the sham group (*Figure 3E*). No significant difference was found on day 3 and 4 (all $t_{(17)} > -1.648$, $p = $ n.s.)

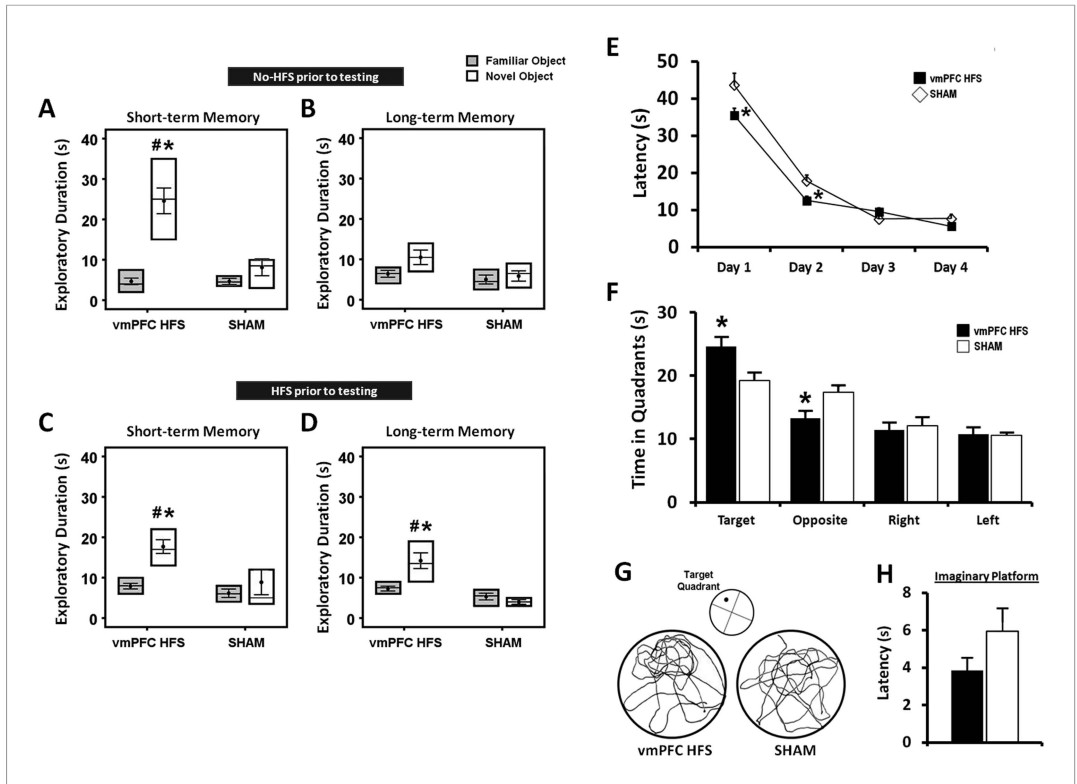

**Figure 3**. Effects of chronic stimulation on the short- and long-term memory retention interval in the novel-object recognition test and Morris water-maze task. Animals were tested in two different conditions with either no-HFS (**A**, **B**) or HFS (**C**, **D**) prior to the task. Effects of chronic stimulation on the memory performance in the Morris water-maze test (**E**, **F**). Note: VmPFC HFS significantly enhanced the short- and long-term memory performances in the novel-object recognition test (**C**, **D**). In the Morris water-maze experiment, there was an improvement on learning and memory after vmPFC HFS in both the training (**E**) and probe test (**F**) phases. Representative swimming paths (**G**) in the probe test, demonstrating vmPFC HFS increased duration within the virtual zone around the platform's location. Latency to reach the imaginary platform showed no difference between the vmPFC HFS and sham animals during the probe test (**H**). Indication: *, significant difference from the sham rats, (p < 0.05); #, significant from the familiar object of respective stimulation amplitude, (p < 0.05).

as all rats eventually learned the task. In the probe test, vmPFC HFS rats spent more time in the target quadrant ($t_{(17)} = 2.511$, $p = 0.022$) and lesser time in the opposite quadrant ($t_{(17)} = -2.713$, $p = 0.015$), suggesting that treated rats had stronger spatial memory for the target quadrant (*Figure 3F,G*). No significant difference was demonstrated in other equivalent quadrants (all $t_{(17)} > -0.386$, $p = $ n.s.). When analyzing the latency to reach the imaginary platform during the probe test, no significant effect was found between the vmPFC HFS and sham animals ($t_{(16)} = -1.505$, $p = $ n.s.; *Figure 3H*).

## Chronic stimulation upregulated neurogenesis-associated genes

In this study, expression of neurogenesis-related genes was quantified using qPCR assay. The selection of candidate genes for qPCR was based on our previous microarray data (*Kadar et al., 2011*). We found significant effects for group [$F_{(1,10)} = 52.948$, $p < 0.001$], genes [$F_{(8,80)} = 386.955$, $p < 0.001$], and the interaction group × genes [$F_{(8,80)} = 2.443$, $p = 0.02$]. Remarkably, vmPFC HFS upregulated genes related with neurogenesis and neuroplasticity (NeuN/Rbfox3, $t_{(10)} = -7.018$, $p < 0.001$; Syn, $t_{(10)} = -4.660$, $p = 0.001$; Dcx, $t_{(10)} = -2.860$, $p = 0.012$; Nes, $t_{(10)} = -3.214$, $p = 0.009$), genes related with neuronal differentiation (Angpt2, $t_{(10)} = -3.520$, $p = 0.006$; and S100a4, $t_{(10)} = -3.372$, $p = 0.007$), as well as genes related with migration and neuroprotective functions (Angpt2: $t_{(10)} = -3.520$, $p = 0.006$) in the hippocampus (*Figure 4A*). No changes were found for Timp1, Ccl2, and BDNF (all $t_{(10)} > -1.781$, $p = $ n.s.). Calculation for the fold-change values using

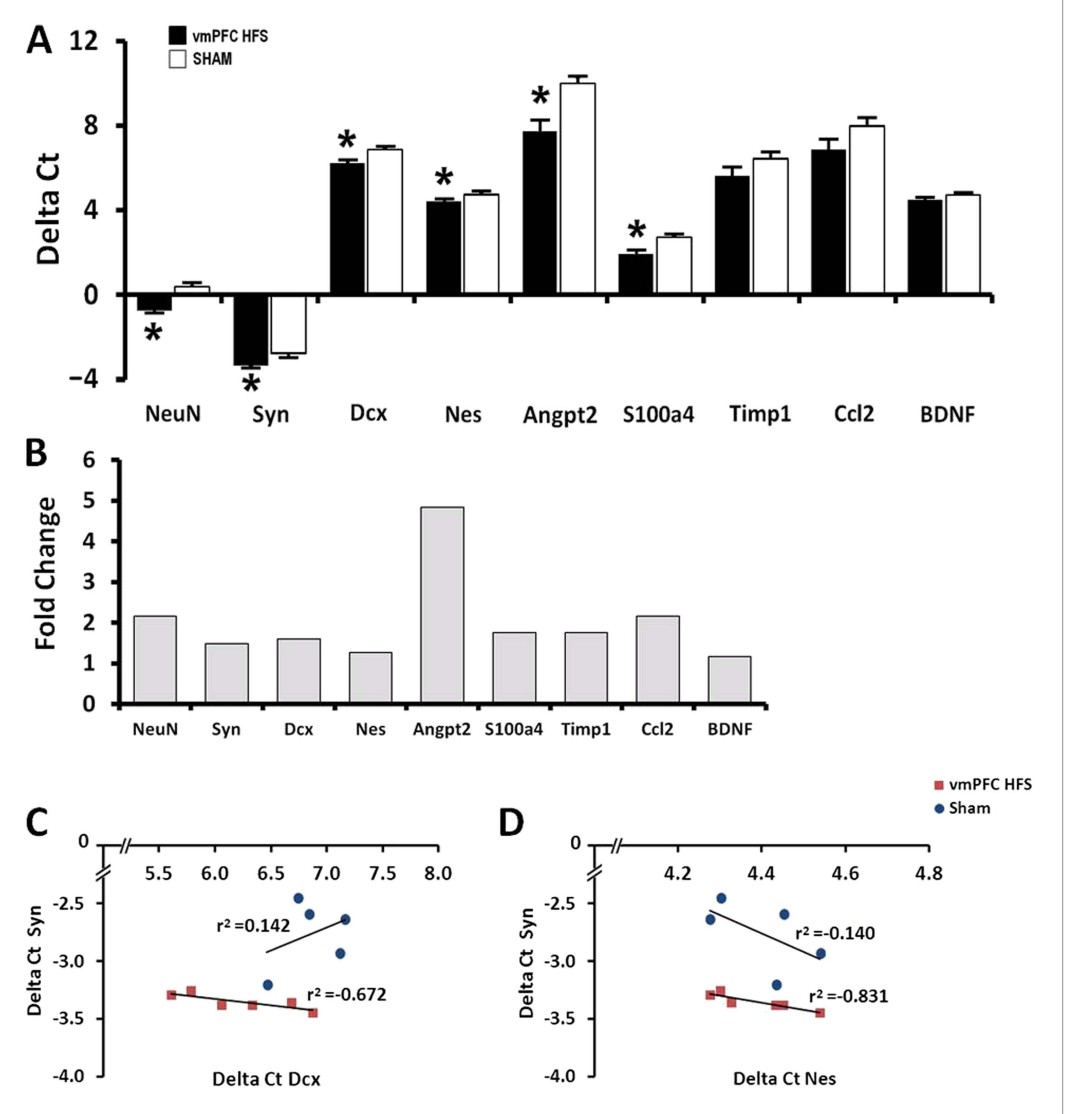

**Figure 4**. Effects of chronic vmPFC HFS on the mRNA gene expression related to neuroplasticity in the hippocampus (**A**). Note: vmPFC HFS upregulated genes involved in proliferation and neurogenesis-related functions including the NeuN, Syn, Dcx, Nes, Angpt2, and S100a4 relative to sham. No changes were found for Timp1, Ccl2, and BDNF. Calculation for the fold-change values indicating that vmPFC HFS induced approximately 4.8-fold (Angpt2), 2-fold (NeuN), and >1-fold (Syn, Dcx, Nes, S100a4) increase of gene expression relative to the sham (**B**). Interestingly, scatter plots show significant correlation between the Syn and the Nes/Dcx (**C**, **D**), indicating that these genes are strongly related to each other for neuroplasticity in the hippocampus after chronic vmPFC HFS. Gene expression was expressed as the change in Ct of the gene of interest compared to the sham (Delta Ct); and relative expression was calculated using the comparative CT method with fold change 2-Delta (Delta Ct). Indication: *, significant difference from the sham rats, ($p < 0.05$).

2-Delta (Delta Ct) method indicated that vmPFC HFS induced approximately 4.8-fold (Angpt2), 2-fold (NeuN/Rbfox3), and >1-fold (Syn, Dcx, Nes, S100a4) increase of gene expression relative to the sham (*Figure 4B*). Although no differences were observed for Timp1, Ccl2, and BDNF genes, their fold-changes were increased by approximately 1.8-fold for Timp1, 2-fold for Ccl2, and 1.2-fold for BDNF, respectively. Interestingly, the gene expression for Syn was significantly correlated with the Nes and Dcx genes (all $r^2 > -0.831$, $p < 0.046$; *Figure 4C,D*), indicating a close association between these genes for induction of neuroplasticity in the hippocampus after chronic vmPFC HFS.

## Stimulation-induced changes from cell proliferation to dendritic spines modification

Given the data of real-time qPCR in which vmPFC HFS induced upregulation of genes associated with neuroplasticity function, we further investigated the stimulation effects on the hippocampal neuronal activity and its morphological changes related to neurogenesis. Our results revealed that vmPFC HFS increased the number of c-Fos-ir positive cells in the subiculum ($t_{(10)} = 2.239$, p = 0.049), DG ($t_{(10)} = 2.992$, p = 0.015), along with a marginal difference in the CA1 field ($t_{(10)} = 2.131$, p = 0.059) of the hippocampus as compared to the non-stimulated sham (*Figure 5A,B*). Although no effect was shown in the CA3 field of the hippocampus ($t_{(10)} = 1.610$, p = n.s.), there was a trend of increased c-Fos-ir neuronal activity after vmPFC stimulation. During correlational analysis of the hippocampal neuronal activity, the c-Fos-ir in the DG was positively correlated with both the hippocampal SUB and CA1 field (all $r^2 > 0.810$, p < 0.015), indicating a strong relationship of the DG with the SUB and CA1 regions after vmPFC HFS (*Figure 5C,D*). No significant correlation between the neuronal activity of c-Fos-ir was found in the hippocampus for sham animals (all $r^2 > 0.025$, p = n.s.).

For morphological study of neurogenesis effects, the DG of the hippocampus was examined by incorporation of BrdU in the DNA of proliferating cells and immunohistochemical detection of BrdU. Our results showed a significant increase of surviving BrdU-positive cells after chronic vmPFC HFS as compared to the sham ($t_{(14)} = 4.371$, p = 0.001; *Figure 5E,F*). Moreover, there was also a remarkable increase in the proliferation of the neural progenitor cells following chronic vmPFC HFS as demonstrated by an early postmitotic neuronal marker Dcx ($t_{(10)} = 4.312$, p = 0.002; *Figure 5G,H*). Interestingly, we found a significant positive correlation between the BrdU and Dcx cell count in the vmPFC HFS ($r^2 = 0.728$, p = 0.031) (*Figure 5I*). Further, after chronic vmPFC HFS, our data demonstrated an increase of dendritic spine density in the secondary (Z = −2.121, p = 0.034), but not the primary branch (Z = −1.061, p = n.s.) of the Golgi-impregnated cells in the DG area of the hippocampus as compared to the sham (*Figure 6A,B*). In comparison between the dendritic spines of the primary and secondary branch, there was a marginal increase of spine density found in the secondary branch of the vmPFC HFS (Z = −1.826, p = 0.068), but not in the sham (Z = −1.604, p = n.s.) animals. To further investigate the effects of DBS-induced neurogenesis increase for memory function, qualitative evaluation of the immunofluorescence double-labeling showed co-localization of c-Fos with the Dcx and the BrdU labeled cells in the hippocampal DG region of the vmPFC HFS animals (*Figure 6C*), indicating a possibility of a substantial role of these newborn cells in the memory enhancement function.

## Correlations between the hippocampal neuroplasticity and the behavioral performances

When correlational study was performed independently for the vmPFC HFS, there was an association between the cell count for BrdU and Dcx after chronic vmPFC HFS ($r^2 = 0.728$, p = 0.031) (*Figure 5I*). To examine the relationship between the neuroplasticity variables and the memory performances, vmPFC HFS induced a significant positive correlation between the MWM target quadrant and the Dcx cell count ($r^2 = 0.672$, p = 0.046; *Figure 7A*). Interestingly, after chronic vmPFC HFS, there was also a positive correlation for the short-term memory in condition of no-HFS prior to the NOR testing with the Dcx cell count ($r^2 = 0.674$, p = 0.045) and NeuN/Rbfox3 gene expression ($r^2 = 0.834$, p = 0.011) (*Figure 7C,D*).

## Discussion

The present findings confirm the results of previous studies of progressive age-related memory impairment in the middle-aged rats (*Rex et al., 2005*; *Kaczorowski and Disterhoft, 2009*). We next conducted electrical stimulation in this animal model of memory deficit, using HFS and LFS with various stimulation current intensities in the NOR test. Our results showed that HFS at 200 µA and LFS at 50, 200, and 400 µA significantly enhanced the memory functions in the short-, but not the long-term memory retention interval when compared to the sham. We next carried out chronic stimulation in this middle-aged rat model. We hypothesized that chronic stimulation would increase both the short- and long-term memory functions via a mechanism of enhanced hippocampal neuroplasticity. Previous studies have shown that memory deficits were partly due to the disruption of the hippocampal neuroplasticity (*Deupree et al., 1993*; *Rex et al., 2005*). Therefore, our hypothesis was driven by findings that DBS in various brain targets increased BDNF level (*Hamani et al., 2012*; *Ying et al., 2012*) and enhanced neurogenesis-related functions (*Toda et al., 2008*; *Kadar et al., 2011*; *Stone et al., 2011*) in the hippocampus. Further, this hypothesis was also supported by the fact that the increased

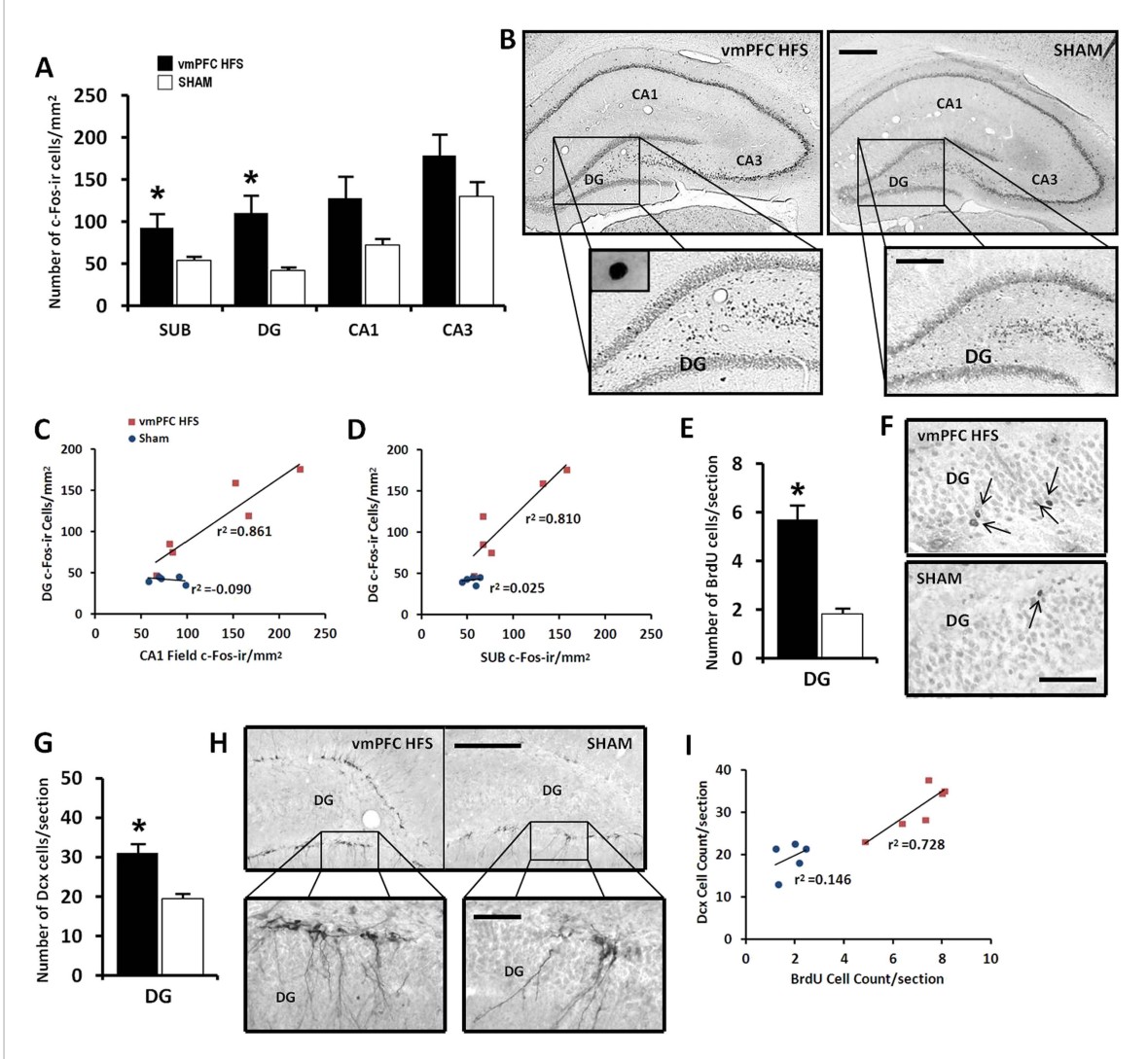

**Figure 5.** Effects of chronic vmPFC HFS on the hippocampal neuronal activity by c-Fos-ir (**A–B**) and the morphological changes related to neurogenesis functions (**E–I**). Note: VmPFC HFS increased the number of c-Fos-ir positive cells in the subiculum, DG, and a marginal difference (p = 0.059) in the CA1 field of the hippocampus as compared to the sham (**A–B**; scale bars: 500 µm, low-power magnification; 250 µm, high-power magnification). In neurogenesis-related morphology, after chronic vmPFC HFS, there was an increase of surviving BrdU-positive cells (**E–F**, scale bar: 500 µm), and neural progenitors—Dcx-positive cells (**G–H**; scale bars: 300 µm, low-power magnification; 50 µm, high-power magnification). For correlational analysis, there was a strong relationship between the BrdU and Dcx cell-count (**I**). Importantly, the neurogenic zone of the DG was also highly correlated with the SUB and CA1 field of the hippocampus (**C–D**), indicating that these regions were functionally associated with memory functions after chronic vmPFC HFS. Abbreviations: SUB, subiculum; DG, dentate gyrus; CA1, CA1 field of the hippocampus; CA3, CA3 field of the hippocampus; vmPFC HFS, high-frequency stimulation of the ventromedial prefrontal cortex; BrdU, 5-bromo-2′-deoxyuridine; and Dcx, doublecourtin. Indication: *, significant difference from the sham rats, (p < 0.05).

BDNF level and neurogenesis function in the hippocampus were strongly correlated with the hippocampal-dependent memory tests (*Drapeau et al., 2003*; *Erickson et al., 2011*). Thus, after chronic stimulation, we found significant improvement in both the short- and long-term memories in the NOR test, as well as the spatial memory performances during the MWM task as compared to the sham.

In the acute stimulation experiment, we found that the behavioral effects of vmPFC stimulation were dependent on the stimulation frequency and current intensity. Although these findings are consistent with previous reports (*Hamani et al., 2010a*; *Hescham et al., 2013b*), the mechanisms of the stimulation parameter dependency in regulation of memory functions remain largely obscure. Nonetheless, it is postulated that the neurons in the vmPFC are highly sensitive to specific stimulation

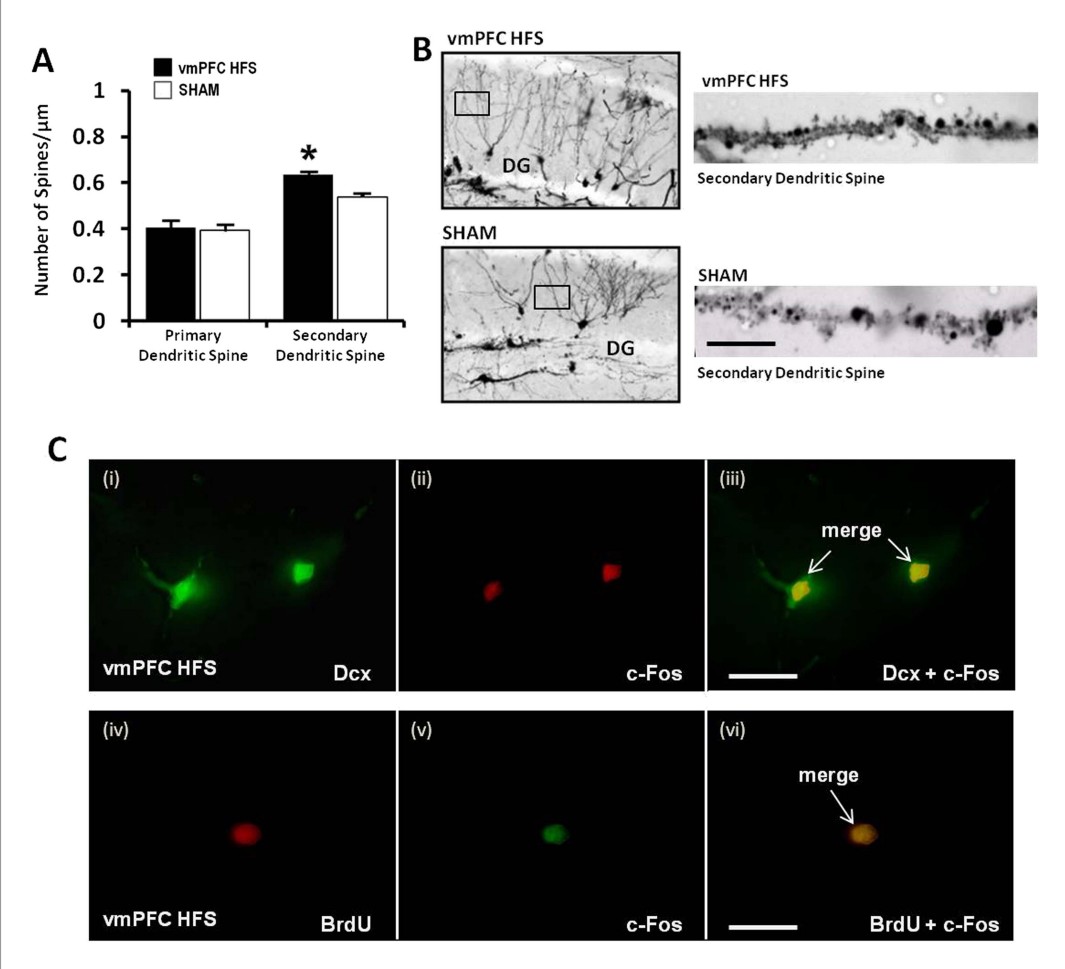

**Figure 6**. Effects of chronic vmPFC HFS on the Golgi measurement of dendritic spine density and immunofluorescence labeling of neurogenesis-related cell function in the hippocampal DG area. Note: there was an increase in the secondary, but not the primary dendritic spine density of the Golgi-impregnated cells in the DG area of the hippocampus (**A–B**; scale bar: 10 μm). Representative confocal images (**C**) are demonstrated for the localization of Dcx (green, **C-i**), c-Fos (red, **C-ii**; or green, **C-v**), and BrdU-labeled (red, **C-iv**) immunofluorescence positive cells. Merged images showed the co-localization of Dcx and c-Fos (**C-iii**), as well as BrdU and c-Fos (**C-vi**) in the hippocampal DG region (scale bar: 40 μm). Indication: *, significant difference from the sham rats, (p < 0.05).

settings for induction of either long-term potentiation (LTP) or depression (LTD). This notion was highly supported by the findings in which the vmPFC synapses are potentially vulnerable to LTP or LTD strengthening after specific type of stimulation (*Herry and Garcia, 2002*). It has been found that the prefrontal cortex neurons could undergo LTP with rapid and stable potentiation in the prelimbic synapses after high-frequency tetanic stimulation of the hippocampal CA1/subicular region (*Laroche et al., 1990*). Such a LTP-like plasticity in the vmPFC after hippocampal HFS has been shown to be dependent on α-amino-3-hydroxy-5-methyl-4-isoxazole propionic acid (AMPA) and N-methyl-D-aspartate (NMDA)-receptor, which indicates a crucial role for the synaptic potentiation in the hippocampal-vmPFC pathway in rapid memory consolidation (*Laroche et al., 2000*). Although HFS and LFS at specific amplitudes were effective for memory enhancement, it is noteworthy that LFS induced negative effects on the anxiety-related behaviors (*Lim et al., 2015b*). Furthermore, HFS of the vmPFC has been shown to reduce conditioned fear and enhance the extinction of aversive memory (*Milad and Quirk, 2002*; *Milad et al., 2004*), in contrast to LFS, which induced impairment in extinction of conditioned fear (*Shehadi and Maroun, 2012*). Since previous

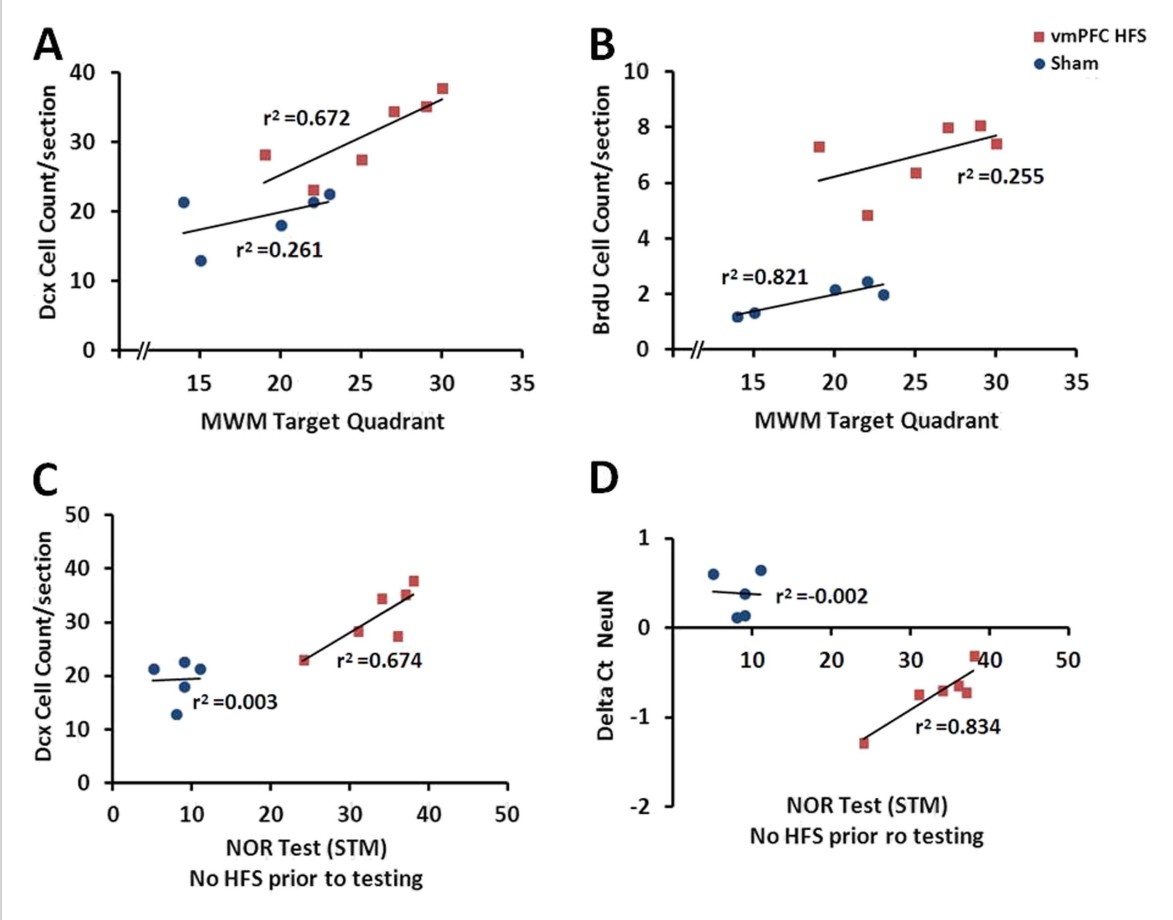

**Figure 7**. Scatter plots display the correlations between the variables related with the hippocampal neuroplasticity and the hippocampal-dependent memory behavioral tests. Note: In the vmPFC HFS animals, the Morris water-maze target quadrant is positively correlated with the Dcx cell count (**A**), while the novel-object recognition task with no-HFS prior to testing shows positive correlation with the Dcx cell count (**C**) as well as the NeuN/Rbfox3 gene expression (**D**). No correlational association was found between the Morris water-maze and BrdU cell count after chronic vmPFC high-frequency stimulation (**B**).

studies of vmPFC HFS produced robust antidepressant-like behaviors (*Lim et al., 2015a*, *2015b*), we therefore applied this stimulation parameter in the chronic stimulation experiment to test the hypothesis that it would restore the memory deficits for both the short- and long-term memory functions of the middle-aged rats.

As might be expected for the effects after chronic stimulation, there was an improvement on memory function in the short- but not the long-term memory retention interval when the animals were tested with no-HFS prior to the NOR task. Strikingly, we found a remarkable reversal of memory deficits in both the short- and long-term memory of the animals that received HFS prior to the NOR test as compared to the non-stimulated sham. Congruently, chronic vmPFC HFS significantly improved the spatial navigation performances when conducted in a hippocampal-dependent memory MWM test. In the present data, we clearly distinguished a role for the vmPFC in regulation of both the short- and long-term memory functions. In most studies, short-term memory is generally regarded as memory spanning from seconds to several minutes or hours, while long-term memory is usually last from hours to several days or longer (*Nagai et al., 2007*; *Euston et al., 2012*). Albeit many studies have indicated that vmPFC is involved in the expression of long-term memory (*Maviel et al., 2004*; *Teixeira et al., 2006*), there are also findings supporting that vmPFC is involved in consolidation and retrieval of recently acquired memories (*Blum et al., 2006*; *Leon et al., 2010*). Thus, it is likely that the vmPFC HFS potentiated the initial hippocampal encoding

during the acquisition phase, which was then followed by enhanced retrieval during the short- and long-term memory recalls. Taken together, these findings further implicate a specific role for the vmPFC HFS in facilitation of rapid consolidation and retrieval of the short- and long-term memory processes in the hippocampus.

HFS of the vmPFC has not only been demonstrated for memory enhancement in the middle-aged memory deficit rat model, but also induced profound antidepressant-like behaviors in the experimental animal studies (*Lim et al., 2015a*; *2015b*). Interestingly, the effects of memory improvement by vmPFC HFS, as characterized by the increased novelty seeking in the NOR and MWM tasks, were highly associated with the stimulation effects on reduction in anxiety behavior. These results confirm previous findings that animals with lower anxiety or fear level displayed higher novelty seeking behavior, particularly in the elevated plus-maze and light–dark box tests (*Kabbaj et al., 2000*; *Stead et al., 2006*). In consistent with the data found in the animal model of depression, rats undergoing chronic stress exhibited anxious and low exploratory behaviors (*Lim et al., 2015b*), as well as impairment in spatial memory task performances (*Conrad et al., 1996*; *Beck and Luine, 1999*). Apparently, chronic stress produces pathological alterations on the molecular and morphological levels in the hippocampus that is involved in the regulation of both the spatial memory formation and emotional behaviors. Thus, our experiments found that the memory enhancement effect is likely to be accompanied by the vmPFC DBS-induced anxiolytic effects via the mechanisms of neurogenesis and dendritic remodeling in the hippocampal neurons.

Given the prominent anatomical connectivity between the hippocampus and vmPFC (*Jay and Witter, 1991*; *Cenquizca and Swanson, 2007*), it was observed that vmPFC HFS drove the local neural activity as characterized by c-Fos-ir activation in the subiculum, DG, as well as a marginal increase in the CA1 field of the hippocampus. Although there is no direct connection from vmPFC to the hippocampus, it is possible that the effects of vmPFC HFS on the hippocampal neural activity are mediated by a reciprocal bisynaptic pathway through the nucleus reunions or the lateral entorhinal cortex (*Burwell and Amaral, 1998*; *Vertes et al., 2007*). Besides, the neural activity in the hippocampus could probably be activated by either antidromic or orthodromic stimulation that possibly achieved by a current spread from the vmPFC structure to the neighboring axon bundles—the minor forceps of the corpus callosum (*Lim et al., 2015b*). As a result, the hippocampal regions would eventually be activated for an induction of LTP to strengthen its synaptic plasticity for memory processes. Of particular interest, a tractography analysis by diffusion tensor imaging provides evidence for this structural connectivity that HFS of the subgenual cingulate gyrus (generally considered to be homologue of the rat vmPFC), showed connections to the medial frontal cortex, anterior and posterior cingulate, and the anterior medial temporal lobe (i.e., amygdala-hippocampus) (*Gutman et al., 2009*). The connections of the hippocampus and amygdala to the vmPFC have been previously investigated by anatomical and electrophysiological studies (*Laroche et al., 1990*; *Jay and Witter, 1991*). It has been shown that the excitatory and inhibitory inputs from the amygdala and hippocampus were converged and interact in the vmPFC (*Ishikawa and Nakamura, 2003*), implying that activation of the amygdalar-hippocampal neurons might be crucial for vmPFC neurons in memory regulation. More importantly, the electrophysiological studies have provided concrete evidence of functional interaction between the vmPFC and hippocampus in which their theta oscillations were highly synchronized as measured by both the spike-theta phase locking and local field potential coherence, during memory acquisition and retrieval in spatial tasks (*Jones and Wilson, 2005*; *Siapas et al., 2005*; *Benchenane et al., 2010*). Based on this evidence, we suggested that the rapid encoding and retrieval of memory depend largely on the bidirectional regulation of synaptic connectivity between the vmPFC and hippocampus; while the disruption of its connection affect the learning and memory functions, which are commonly identified in patients suffering from dementia (*Salat et al., 2001*).

In line with our findings, HFS of the anterior nucleus of the thalamus and the entorhinal cortex has been demonstrated to increase neurogenesis (*Toda et al., 2008*) and memory functions, particularly spatial memory measured in the MWM (*Stone et al., 2011*) and enhanced performance on a delayed non-matching to sample task (*Hamani et al., 2011*). In this study, we found that vmPFC HFS induced upregulation of neurogenesis-associated genes with increased neural progenitors cells and dendritic spines in the DG of the hippocampus. In agreement with our previous microarray data (*Kadar et al., 2011*), chronic HFS has been shown to modulate the hippocampal genes that involved in the proliferation and neurogenesis-related functions, as well as genes supporting for neural differentiation,

migration and maturation. Although we observed no differences for the Timp1 (neuroprotection), Ccl2 (neural differentiation), and BDNF (synaptic plasticity) gene expression, there was an overall increase in their fold-change with approximately 1.2–2 fold after chronic vmPFC HFS. Notably, we found a strong induction of upregulation in the Angpt2 gene (4.8-fold) that promotes neuronal differentiation, migration and neuroprotection (*Liu et al., 2009*), as well as NeuN/Rbfox3 gene (2-fold), which plays an essential role for neural progenitor cells differentiation and maturation (*Kim et al., 2009*, *2013*). Further, our present observations were also supported by earlier works demonstrating that vmPFC HFS induced significant increase of BDNF and serotonin (5-HT) levels in the hippocampus (*Hamani et al., 2012*). Recent studies have shown that the increase of 5-HT and BDNF expression in the hippocampus regulate synaptic plasticity, as well as cognitive and mood-related behaviors. It is well-known that both the 5-HT and BDNF promote neurogenesis by enhancing the synaptogenesis, neuronal differentiation, and survival particularly for memory acquisition and consolidation (*Gaspar et al., 2003*; *Lu and Chang, 2004*; *Pang et al., 2004*). Although no microdialysis data were provided with regard to the extracellular levels of 5-HT and BDNF after chronic vmPFC HFS, the increased neurogenesis effects as obtained from this study clearly indicate that its plausible mechanism is facilitated by the release of 5-HT and enhanced BDNF levels (*Hamani et al., 2012*). Importantly, the vmPFC HFS effects on the behavioral observation of improved memory functions and the neurogenesis have indicated that a strong synaptic network circuitry has been established within the DG for integration of new information and memory storage. In support of this notion, previous studies have demonstrated that the increased neurogenesis was highly associated with enhanced learning and memory, while its decrease caused memory impairment (*van Praag et al., 1999*; *Shors et al., 2001*). Although the present study has identified the DBS-induced memory enhancement by neurogenesis, there is a possibility that these effects are mediated by other non-neurogenic mechanisms such as modulation of the neurotransmission (via acetylcholine, dopamine, 5-HT, etc), synaptic potentiation by the AMPA/NMDA receptor trafficking, and enhancement of the BDNF or CREB (cAMP response element-binding protein) function (*Hamani et al., 2012*; *Stern and Alberni, 2012*).

The above findings prompted us to further examine whether the increase of neurogenesis-related functions is associated with the effects of memory enhancement by vmPFC HFS. Our correlation analysis of gene expression revealed that the Syn was strongly associated with the Nes and Dcx genes, suggesting that vmPFC HFS might possibly cause an alteration of increased dendritic synaptogenesis, particularly in the hippocampal DG where transcriptional process for Nes and Dcx genes occurs. Consistent with this interpretation, we have demonstrated the increase of dendritic spines in the Golgi-impregnated cells of the hippocampal DG. This observation was supported by a previous study which showed that electrical stimulation affected the axonal path in cultured Xenopus neurons that was mediated by elevation of both cytoplasmic $Ca^{2+}$ and cyclic adenosine monophosphate levels (*Ming et al., 2001*). Importantly, our behavioral correlates show that the spatial memory performances were associated with the cell proliferation in the DG, indicating that neurogenesis in the DG is vital for the hippocampal-dependent learning and memory processes.

In conclusion, our findings suggest that chronic vmPFC HFS induces long-lasting effects on memory performances and its underlying mechanism is possibly mediated by an enhanced neurogenesis in the hippocampus. Despite the fact that this structure has been previously shown for antidepressant-like activities, it might as well serve as a new effective DBS target for aged-related memory deficits. Thus, this translational research provides an additional window for a possibility of future clinical trials on this potential brain target for memory enhancement.

## Materials and methods

### Subjects

Male Sprague–Dawley rats (12 month old, n = 144; and 4 month-old, n = 20; National University of Singapore, Singapore) were individually housed with ad libitum access to food and water. The animal colony was maintained under controlled temperature (about 24–26°C), humidity (60–70%), and 12 hr dark/light cycle (lights-off at 0700). All procedures were approved by the Institutional of Animals Care and Use Committee of Nanyang Technological University.

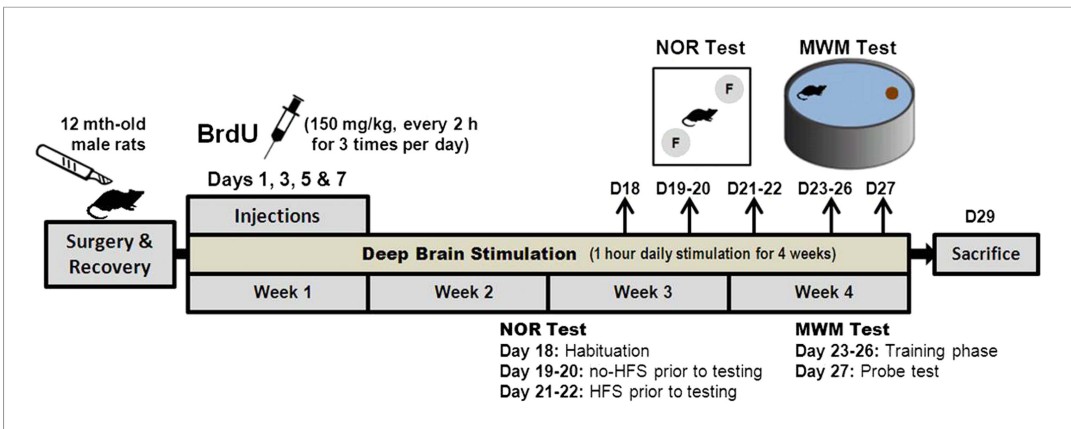

**Figure 8**. Schematic representation of the experimental design for chronic stimulation and behavioral testing of memory functions in the middle-aged animals.

## Experimental design

The NOR test was performed to compare the short- and long-term memory functions between the young (n = 20) and middle-aged (n = 15) rats (*Figure 1*). In the acute DBS experiments, animals (n = 102) received either HFS (n = 40) or LFS (n = 46), at various stimulation amplitudes (n = 10–12 per group) (*Figure 2*). The sham group (n = 16) was similarly operated with electrode implantation in the vmPFC. In the chronic DBS experiment, animals (vmPFC HFS, n = 15; sham, n = 12) received stimulation 1 hr daily for a period of 4 weeks. In week 1, animals were intraperitoneally injected with 5-bromo-2′-deoxyuridine (BrdU, Sigma, Missouri, USA; 150 mg/kg per injection; at 2-hr intervals for three times per day) on day 1, 3, 5, and 7 (*Figure 8*). The first injection dose was performed immediately before the 1-hr stimulation, followed by the second and third dose at 2-hr intervals. The memory functions were tested using the hippocampal-dependent memory tasks—NOR and the MWM tests on days 18–27.

## Deep brain stimulation

Surgery was performed as previously described (*Lim et al., 2010, 2015a*). In brief, rats were anesthetized (2.5% isoflurane inhalation) and placed in a stereotactic frame (Vernier Stereotaxic Instrument, Leica Biosystems, Nussloch, Germany). A bilateral stimulating electrode was implanted in the vmPFC (AP: +2.70 mm; L: ±0.60 mm; V: 4.60 mm) based on the rat brain atlas of *Paxinos and Watson (1998)*. All animals were given a 2-week recovery period.

For stimulation, a bipolar stimulating electrode (Synergy, Singapore) was constructed using an inner platinum-iridium core wire with a gold-plated cannula (Technomed, Beek, Netherlands) (*Lim et al., 2009*; *Tan et al., 2010*). A digital stimulator DS8000 and stimulus isolators DLS100 (World Precision Instruments, Sarasota, USA) were used to deliver the electrical stimuli. In the acute DBS experiment, either HFS (100 Hz) or LFS (10 Hz) with stimulation amplitudes of 50, 100, 200, and 400 µA was used. The pulse width was set at 100 µs. In the chronic DBS experiment, the stimulation parameter (HFS, 200 µA amplitude, and 100 µs pulse width), derived from the present acute DBS study (*Figure 2*) and previous findings (*Hamani et al., 2010a*; *Lim et al., 2015b*), was used to test the hypothesis that chronic stimulation enhances both the short- and long-term memory functions.

## Behavioral testing

In behavioral experiments, all animals in the acute and chronic stimulation studies received electrical stimulation for approximately 30 min immediately prior to each of the NOR test phases, and MWM training and probe tests. The behavioral testing was conducted in a dimly lighted room from 8:00 till 14:00 hr. In the chronic stimulation experiments, all animals were stimulated daily for 1 hr during 14:00–19:00 hr. However, animals that had already received the 30-min stimulation prior to the behavioral testing were again stimulated for another 30 min during 14:00–19:00 hr, so that each animal received a total of

1-hr stimulation per day. Sham animals were similarly connected to cables, without electrical stimulation. Animals were handled daily to avoid unnecessary stress during behavioral experiments.

## Novel-object recognition test

The animal was habituated in an empty open-field (100 × 100 × 40 cm) arena for a 20-min exploration. The next day, the animal was exposed to two identical objects placed in the open-field during the acquisition phase for 3 min. 90 min after acquisition phase, the animal was exposed to the same arena and presented with a familiar and a novel object for 3 min, and subsequently tested for the short-term memory. After 24 hr following the short-term memory retention interval, the animal was again exposed to the same arena and presented with a familiar and another novel object for 3 min, to test for the long-term memory function (*Figure 1A*). The positions of the familiar and novel objects were counter-balanced across tests and subjects. After each trial, the objects and the open-field were cleaned with 70% ethanol to minimize olfactory cues. All behavior was video recorded for offline analysis.

In the chronic experiments, the animals were tested with either no-HFS or HFS prior to the behavioral testing from day 19 to day 22. On day 18, the animal was habituated in an empty open-field arena for 20 min exploration. In no-HFS condition (day 19–20), the animals were connected to the cable without electrical stimulation for 30 min before each of the NOR test phase. In HFS condition (day 21–22), the animals received electrical stimulation for 30 min and immediately tested in the NOR task. For event sequence of all behavioral procedures in the chronic experiment, see *Figure 8*.

## Morris water-maze test

The apparatus consists of a black circular pool (128 cm diameter, 60 cm high), filled with water (30 cm depth) maintained within 23–25°C. The pool was spatially divided into four imaginary quadrants: target, opposite, left, and right. A circular, transparent escape platform (10 cm diameter) was placed 2 cm below the water surface in the target quadrant of the pool. In the training phase, the rats were trained to locate the submerged platform for 4 consecutive days with 4 trials per day (Trial duration: 2 min, inter-trial interval: 1 min). The starting positions were randomized, but all were equidistant from the platform. If the rats failed to locate the platform in 1 min, then they were gently guided to the platform. The escape latency was measured on each trial. The probe test was conducted 24 hr after the final training trial with removal of platform from the pool. The animals were allowed to swim for 60 s and the duration spent in each quadrant was video recorded for offline analysis.

## Histological processing

2 days after the last behavioral test, the animals were stimulated for 2 hr and immediately decapitated with isoflurane anesthesia. The brains were subsequently removed, frozen in liquid nitrogen, and stored at −80°C for gene expression and morphological studies. For Golgi study, the brains (vmPFC HFS, n = 4; Sham, n = 3) from the chronic stimulation experiment were removed after decapitation, processed for rapid Golgi staining, and coronal sections of 100 µm were obtained as previously described (*Vyas et al., 2003*). For immunohistochemistry, the brains (vmPFC HFS, n = 8; Sham, n = 6) from the chronic stimulation experiment were serially cut on a cryostat CM3050 (Leica Microsystems, Wetzlar, Germany) into 20-µm coronal sections and collected on gelatin-coated slides. For neurogenesis-related gene expression study, the hippocampal region from the chronic stimulation experiment was separately collected in Eppendorf tubes and micro-dissected (400 µm thickness) for real-time quantitative polymerase chain reaction (qPCR) assay.

Before staining, all sections were incubated with 4% paraformaldehyde for 1 hr, followed by 0.3% $H_2O_2$ treatment for 10 min. For BrdU staining, sections were first incubated in 2N HCl for 30 min at 37°C. The primary antibody incubation was performed using a rabbit anti-c-Fos antibody (diluted 1:400), mouse-anti-BrdU antibody (diluted 1:50), and goat anti-Dcx antibody (diluted 1:50) (all primary antibodies, Santa Cruz Biotechnology, Inc, Dallas, USA), for 3 days at 4°C. After rinsing, all sections were incubated with a corresponding secondary biotinylated goat anti-rabbit antibody, biotinylated horse anti-mouse antibody, or biotinylated rabbit anti-goat antibody (all dilutions 1:200; Vector Laboratories, Burlingame, CA, USA) for 1 day. Next, the sections were incubated with an avidin-biotin-peroxidase complex (diluted 1:200 in standard Vectastain Elite ABC kit; Vector Laboratories) for 4 hr, followed by incubation in solution 3,3′-diaminobenzidine tetrahydrochloride (DAB Substrate Kit; Vector Laboratories) with nickel chloride enhancement for visualization of the immune complex of the horseradish peroxide reaction product. Finally, all sections were dehydrated, and cover-slipped with Permount (Thermo Fisher Scientific, Waltham, USA).

For histological quantification, the counting for c-Fos immunoreactive (c-Fos-ir), BrdU and Dcx-positive cells (*Lim et al., 2008*; *Hestermann et al., 2014*), and measurement of dendritic spine density (*Vyas et al., 2003*) were performed using previously established methods with minor modifications. In brief, the c-Fos-ir cells per mm$^2$ (six sections per animals) was counted in the subiculum, dentate gyrus, CA1, and CA3 field of the hippocampus using image analysis program 'Image J' (version 1.47, NIH, USA). For BrdU- and DCX-positive cells, quantification (six sections per animal) was performed using a bright-field microscope (Olympus, Japan). For Golgi measurement of dendritic spine density, the quantifications were restricted to the primary and the secondary dendrite branches (each, six sections per animal). The spines were counted along a 50-μm stretch of the dendrite starting from the origin of the soma or secondary branch at 100× magnification (Olympus BX43 microscope, 100× Objective, Tokyo, Japan).

## Immunofluorescence staining

The immunofluorescence staining was performed based on previously established protocols (*Temel et al., 2012a*; *Lim et al., 2015b*). After pre-blocking for 30 min in PBS-Triton (PBS-T) with 10% normal donkey serum, two double labeling stainings were carried out on the vmPFC HFS hippocampal sections using goat anti-Dcx (1:50) and rabbit anti-c-Fos antibody (1:500); as well as mouse anti-BrdU (1:50) and goat anti-c-Fos (1:100) as primary antibodies (all antibodies, Santa Cruz Biotechnology, Inc.) in 5% normal donkey serum for 3-day incubation. After rinsing, the sections were incubated with corresponding Alexa Fluor secondary antibodies (Alexa Fluor 594 donkey anti-rabbit, Alexa Fluor 488 donkey anti-goat, and Alexa Fluor 594 horse anti-mouse; each 1:200; Vector Laboratories) for 2 hr at room temperature. Finally, the sections were mounted on the Superfrost micro-slides (VWM, Illinois, USA) and cover-slipped with Vectashield (Vector Laboratories). To analyze sections for co-localization of cells, photographs were taken using a digital camera that was connected to a laser-scanning confocal microscope (Carl Zeiss, Oberkochen, Germany).

## Real-time PCR

Total RNA was isolated from the hippocampal area (400 μm) of frozen tissue using TRIzol reagent (Life Technologies, Carlsbad, USA) as recommended and converted into cDNA. Real-time qPCR for neuroplasticity-related gene expression (Neuronal nuclei, NeuN/Rbfox3; Synaptophysin, Syn; Double-courtin, Dcx; Nestin, Nes; Angiopoietin-2, Angpt2; S100-calcium-binding protein a4, S100a4; TIMP metallopeptidase inhibitor-1, Timp1; Chemokine [C–C motif] ligand-2, Ccl2; and BDNF) was performed using thermal cycler (Applied Biosystems 7500, Foster City, USA) and SYBR Green quantitative PCR mix (Applied Biosystems, Life Technologies, Warrington, UK). For details of primer sequences used, see *Table 1*. Data analysis of relative gene expression with reference to internal control by real-time PCR (Delta Ct) was quantified. Fold change was calculated using the comparative CT method as the ratio of the 2$^-$ and the 2(-Delta Delta C(T)) method.

**Table 1.** The primers sequences used for real-time quantitative-PCR analysis

| Gene symbol | 5′–3′ primer sequence |
| --- | --- |
| NeuN (Rbfox3) | Fwd 5′–GGCTGGAAGCTAAACCCTGT–3′; Rev 5′–TCCGATGCTGTAGGTTGCTG–3′ |
| Syn | Fwd 5′–GTGCCAACAAGACGGAGAGT–3′; Rev 5′–TTGGTAGTGCCCCCTTTGAC–3′ |
| Dcx | Fwd 5′–ACGACCAAGACGCAAATGGA–3′; Rev 5′–AGGCCAAGGATCTGACTTG –3′ |
| Nes | Fwd 5′–TAAGTTCCAGCTGGCTGTGG–3′; Rev 5′–ATAGGTGGGATGGGAGTGCT–3′ |
| Angpt2 | Fwd 5′–GGACCCTGCAGCTACACATT–3′; Rev 5′–TGTCACAGTAGGCCTTGACC–3′ |
| S100a4 | Fwd 5′–CTTGGTCTGGTCTCAACGGT–3′; Rev 5′–GCAGCTTCGTCTGTCCTTCT–3′ |
| Timp1 | Fwd 5′–ACGCTAGAGCAGATACCACG–3′; Rev 5′– GATCGCTCTGGTAGCCCTTC–3′ |
| Ccl2 | Fwd 5′–AGCCAACTCTCACTGAAGCC–3′; Rev 5′–TGGGGCATTAACTGCATCTGG–3′ |
| BDNF | Fwd 5′–AGGACAGCAAAGCCACAATGTTC–3′; Rev 5′–TTGCCTTGTCCGTGGACGTTTG–3′ |
| HPRT | Fwd 5′–AGGCCAGACTTTGTTGGATT–3′; Rev 5′–GCTTTTCCACTTTCGCTGAT–3′ |

## Statistical analysis

Data analysis was performed using the IBM SPSS Statistics 20. The results were presented in box plots (with interquartile ranges and S.E.M) or mean $\pm$ S.E.M, unless otherwise indicated. Kolmogorov–Smirnov test was used to examine the data normality distribution. The behavior data were analyzed by either three-way or four-way ANOVA with repeated-measures, and Bonferroni post-hoc tests or independent sample $t$-test was used for detailed comparisons, as appropriate. Paired sample $t$-test was used to compare differences between the novel and familiar objects. The data for gene expression and morphological study were analyzed by either independent sample $t$-test or non-parametric Mann–Whitney U test, as appropriate. Pearson correlation coefficients with Bonferroni correction were calculated to examine the relationship between different variables related with the hippocampal neuroplasticity and the behavioral measures. All p-values $<0.05$ were considered significant.

## Acknowledgements

The authors thank Sharafuddin Khairuddin for technical support in histochemistry. The scientific work was funded by the Singapore Lee Kuan Yew Research Fellowship (M4080846.080) that awarded to LWL.

## Additional information

### Funding

| Funder | Grant reference | Author |
|---|---|---|
| Nanyang Technological University | Lee Kuan Yew Research Fellowship (M4080846.080) | Lee Wei Lim |

The funder had no role in study design, data collection and interpretation, or the decision to submit the work for publication.

### Author contributions

AL, Analyzed parts of the behavioural and histochemical data; NJ, Conducted parts of the real-time PCR experiments; AV, Contributed to part of the data analysis and comments on the manuscript; LWL, Conception and design, Acquisition of data, Analysis and interpretation of data, Drafting or revising the article, Contributed unpublished essential data or reagents

### Ethics

Animal experimentation: All procedures were approved by the Institutional of Animals Care and Use Committee of Nanyang Technological University, Singapore, with the reference number ARF-SBS/NIE-A 0169 AZ.

## Additional files

### Supplementary files

• Supplementary file 1. The tables show the total exploratory duration during the acquisition phase, short- and long-term memory retention intervals in the novel-object recognition test for animal experiments of comparisons between the young and middle-aged rats (A), acute stimulation (B, C), and chronic stimulation (D, E) studies. Indication: *, significant difference from the sham rats.

• Supplementary file 2. The tables show the total exploratory duration for both identical object 1 and 2 during the acquisition phase for animal experiments of comparisons between the young and middle-aged rats (A), acute stimulation (B), and chronic stimulation (C) studies.

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
