## [Decision Letter]

Thank you for sending your work entitled “Ventromedial prefrontal cortex stimulation enhances memory and hippocampal neurogenesis in the middle-aged rats” for consideration at *eLife*. Your article has been favorably evaluated by a Senior editor, Howard Eichenbaum (Reviewing editor), and three reviewers, two of whom, Paul Frankland and Magdalena Sauvage, have agreed to share their identity.

The Reviewing editor and the other reviewers discussed their comments before we reached this decision, and the Reviewing editor has assembled the following comments to help you prepare a revised submission.

The Reviewing editor and all three reviewers were enthusiastic about this study and cited several strengths, including the hypothesis driven design, parametric study of stimulation parameters, robust behavioral effects, and progress towards identifying at least one mechanism. At the same time, there are major concerns about some of the analyses and about the specific role of neurogenesis. Also, each reviewer had additional substantial concerns about the description of the study and potential addition of an experiment, about the breadth of interpretation of mechanisms, and other issues. These are included below.

Reviewing editor comments:

1) First, the data show the strongest enhancement at one HFS level, but also more consistent results with LFS. Thus the claim that HFS is optimal is really an overstatement, especially because they did not compare HFS and LFS on chronic stimulation, or water maze or cellular measures.

2) Second, it is curious that the loss of discrimination between familiar and novel stimuli in the novel object test is a decrease in exploration time for the novel stimulus more so that an increase in exploration time for the familiar stimulus. Forgetting should be apparent, conversely, in an increase in exploration of familiar stimuli, indicating the animal is treating the old stimuli as new. So, the results seems to be the opposite in that middle aged rats seem to treat the new stimuli as old rather than treat the old stimuli as new (also see also McTighe et al. Science 2010). Given that the animals must distribute their total exploration time between two stimuli, it seems might be too much to expect the animal to have high exploration times for both the new and old stimulus. Nevertheless, if the animals are forgetting, I would expect equal exploration of the old and new stimuli in test and in the original acquisition phase. Was this the case? They should show the acquisition phase data and compare scores between acquisition and test.

Reviewer 1:

The weaker part of the paper is linking the pro-cognitive effects of DBS to a mechanism. They provide lots of correlative evidence that neurogenesis changes, and might be a mechanism. However, presumably many, many things also change following DBS, and establishing that the DBS-induced memory improvements depend on DBS would move the paper into the publishable realm I think. Asking whether preventing the DBS-induced neurogenesis increase prevents the memory improvements has been done in other studies (including the Stone et al. paper cited in the current manuscript), and doing such an experiment here would greatly strengthen this paper.

Reviewer 2:

The description of the NOR study did not indicate whether the position of the new object and the familiar object are counterbalanced (so that the effect cannot be due to the animals preferring a certain location independently of the type of object placed in this location). If this was the case, the authors should describe this in the Material and methods section. If not, it should be discussed in the Discussion why this could not be the case and/or evidence for lack of spatial bias of the animals for the location where the new objects were placed should be provided (e.g. during the habituation, exploration time spent without object in the different quadrants should be measured and the time spent in the quadrant where will be placed the new objects shown not to be significantly different from that spent in other quadrants).

Reviewer 3:

1) There were a few areas where I thought the paper could be made clearer. First, the use of English needs to be improved. Second, more information is needed about how the stimulation was administered. For the acute stimulation study, it appears that stimulation was given for 30min before testing; does that mean the 30 minutes immediately before the habituation phase, or before each of the test phases? If it were the latter, stimulation would presumably affect retrieval processes and not encoding. Also, in the chronic stimulation study, it would be useful to know when the 1h of stimulation was given—at a set time of day? When did the stimulation occur relative to behavioral testing? I also would like to see explained what the no-HFS and HFS conditions referred to- perhaps, this condition refers to whether the stimulation session occurred just before the behavioral testing? Given this was a behaviorally relevant variable it should be spelled out clearly. In short, I do not think the stimulation procedure was described in sufficient detail for replication. Perhaps a figure given the sequence of events of all behavioral procedures would be useful.

2) I felt that the presentation of the correlation data was a bit muddled. I did not see a lot of value in computing an entire correlation matrix on this dataset with a relatively low N. A more useful approach might be to simply explore hypothesis-driven correlations (e.g. neurogenesis measures with behavior). I also do not think it is useful to present correlations with both sham and stimulated animals. If the hypothesis is that the benefit of stimulation is related to the degree to which it enhances neurogenesis, there is no reason to include sham animals.

3) Finally, I had two conceptual issues. First, the study is strongly motivated as addressing memory impairments as a result of age-related dementia. However, the animal model used here (12 month old rats) would seem to be a better model of healthy age-related memory decline. In fact, it seems the hypothesized mechanisms may not be effective when significant neural degeneration is present. Second, it may be worth discussing the relationship between mood effects of chronic ventromedial prefrontal cortex stimulation and the memory effect reported here. It may be that the stimulation has an effect on reducing anxiety, which increases novelty seeking and reduces thigmotaxic behavior in the Morris water maze, both which would enhance performance. The authors may want to consider whether the effect of ventromedial prefrontal stimulation on memory could be mediated by its anxiolytic action.

[Editors' note: further revisions were requested prior to acceptance, as described below.]

Thank you for resubmitting your work entitled “Ventromedial prefrontal cortex stimulation enhances memory and hippocampal neurogenesis in the middle-aged rats” for further consideration at *eLife*. Your revised article has been favorably evaluated by a Senior editor and a member of the Board of Reviewing Editors. The manuscript has been improved but there are some remaining issues that need to be addressed before acceptance, as outlined below:

With respect to the authors’ response to Reviewer 1’s point 3, since the authors were unwilling to test whether blocking the DBS-induced enhancement in neurogenesis would prevent the memory enhancement, I would suggest they add a 1-2 sentence discussion noting that they have not excluded that these effects are mediated by non-neurogenic mechanisms and identify one or two possibilities.

---

## [Author Response]

*1) First, the data show the strongest enhancement at one HFS level, but also more consistent results with LFS. Thus the claim that HFS is optimal is really an overstatement, especially because they did not compare HFS and LFS on chronic stimulation, or water maze or cellular measures*.

Given the editor’s assertion that HFS is an overstatement, we have adapted the statements accordingly. Although HFS and LFS at specific amplitudes were effective for memory enhancement, it is noteworthy that LFS induced negative effects on the anxiety-related behaviors (43). Furthermore, HFS of the vmPFC has been shown to reduce conditioned fear and enhance the extinction of aversive memory (Milad et al., 2002), whereas LFS induced impairment in extinction of conditioned fear (62). Since in previous studies, vmPFC HFS produced robust antidepressant-like behaviors (43), we applied this stimulation parameter (HFS) in the chronic stimulation experiment to test the hypothesis about its effect to restore the memory deficits for both the short- and long-term memory functions of the middle-aged rats. We have incorporated this comment into the Discussion (see second paragraph).

*2) Second, it is curious that the loss of discrimination between familiar and novel stimuli in the novel object test is a decrease in exploration time for the novel stimulus more so that an increase in exploration time for the familiar stimulus. Forgetting should be apparent, conversely, in an increase in exploration of familiar stimuli, indicating the animal is treating the old stimuli as new. So, the results seems to be the opposite in that middle aged rats seem to treat the new stimuli as old rather than treat the old stimuli as new (also see also McTighe et al. Science 2010). Given that the animals must distribute their total exploration time between two stimuli, it seems might be too much to expect the animal to have high exploration times for both the new and old stimulus. Nevertheless, if the animals are forgetting, I would expect equal exploration of the old and new stimuli in test and in the original acquisition phase. Was this the case? They should show the acquisition phase data and compare scores between acquisition and test*.

The editor has raised an intriguing question concerning the discrimination between familiar and novel stimuli in the novel object recognition test. The comments of the reviewer based on a recently published paper by McTighe et al. 2010 suggest that the middle-aged rats were possibly treating the novel item as familiar object, while the repeated object as novel item. Please see below the following three successive points with explanation in respect to the questions raised:

a) There are methodological differences with McTighe et al. 2010. One study by Albasser et al. 2011 demonstrated that rats with perirhinal damage were severely impaired in discriminating a novel object from a simultaneously presented familiar item. The latter study showed a normal exploration level for both the novel and the familiar objects, as opposed to the findings by McTighe et al. 2010, which revealed that rats with perirhinal damage perceive novel objects as familiar or perceive familiar objects as novel.

The discontinuous testing procedure performed in McTighe et al. (2010) study could be another possible factor contributing to the different findings. They conducted discrete study and test phases, separated by removal of the rat into a holding cage/different environment for 1h. In contrast, our testing procedure was continuous without interruption or placing the rats in different environments. In addition, many studies were not in favor of the longer retention period of the tests, particularly involving habituation of rats to objects after lengthy intervals (Mumby et al. 2007; Albasser et al. 2009, 2011).

Importantly, the study by McTighe et al. (2010) was performed with rats having perirhinal cortex damage; while the animal model used in our study was considerably healthy with merely aged-related memory decline (intact brain). Indeed, human and animal studies have shown that damage to the perirhinal cortex could result in severe impairments in object recognition (Winters et al., 2010; Alvarev et al., 1994). In the study of McTighe, brain damage can produce impairments in visual recognition memory tasks not because the repeated object looks novel, but instead the novel object looks familiar. Concerning object information itself, the perirhinal cortex is considered to be the most crucial structure (Winters et al., 2008), and its damage would lead to a reorganization of the whole memory system particularly for specific type of memory and stimulus representations in different brain regions. It is, therefore, notable that the animal model used in our study for memory evaluation appears to be very different from the perirhinal cortex damage model.

b) We have congruent data from MWM showing that effects on memory are consistent. The novel object recognition task has become a widely used model for the investigation into memory alterations. Albeit this task can be configured to measure the working memory and preference for novelty in rodents, validation of the results by other established behavioral tasks is advisable/obligatory. In this regard, we have conducted the Morris water maze test to further support our findings obtained from the novel object recognition test. Note. The results of both the novel object recognition and the Morris water maze tasks have consistently confirmed the memory enhancement effects of chronic vmPFC HFS. Thus, we can possibly rule out the probability of paradoxical false memory as found by McTighe et al. (2010) in the middle-aged animals.

c) New data are added as requested. As requested by the editor, for the acquisition phase data, we have included these results in the manuscript (see [Supplementary-material SD2-data]). During the acquisition phase, no significant difference was obtained between the two identical objects, except for one particular case in Acute Stimulation study with low-frequency stimulation at 400μA, and the effect was marginally significant at p=0.045. Overall, our results clearly demonstrated the equal exploration duration for the familiar and novel objects in all the test phases for sham non-stimulated rats, indicating memory decline in the middle-aged animal model. Finally, all results were compared between the acquisition and test phases (please see the first paragraphs of the Results section)

Reviewer 1:

*The weaker part of the paper is linking the pro-cognitive effects of DBS to a mechanism. They provide lots of correlative evidence that neurogenesis changes, and might be a mechanism. However, presumably many, many things also change following DBS, and establishing that the DBS-induced memory improvements depend on DBS would move the paper into the publishable realm I think. Asking whether preventing the DBS-induced neurogenesis increase prevents the memory improvements has been done in other studies (including the Stone et al. paper cited in the current manuscript), and doing such an experiment here would greatly strengthen this paper*.

We agree with the comment of the reviewer that linking the pro-cognitive effects of DBS to correlative evidence of neurogenesis changes might not necessarily reflect the mechanism of DBS-induced memory improvements in the middle-aged animals. Further, we also agree with the reviewer that conducting an additional experiment of preventing the DBS-induced neurogenesis increase for memory improvement would strengthen the hypothesis of this paper.

However, the relationship of neurogenesis with hippocampal memory has already been well established. After blocking hippocampal neurogenesis, spatial memory deficits have been reported in several studies (Fan et al. 2007; Synder et al. 2005); and indeed preventing loss of neurogenesis reversed memory deficits ([68]; Encinas et al. 2011). Notably, many studies have shown that the newly-generated neurons, once sufficiently mature, are activated and expressed during memory formation (Kee et al. 2007; Tashiro et al. 2007).

In consideration of the time constraint and experimental duration (approximately 2 months for surgery and behavior) as well as the availability of middle-aged animals (12-months animal holding) to perform this study, it would take approximately 2 years to complete the entire experiment.

We recognize the importance of the suggested experiment to prove the effects of DBS-induced neurogenesis increase for memory enhancement, we have conducted alternative experiments of immunofluorescence double-labeling of c-Fos (one of the immediate early gene family markers for neuronal activity) with the Dcx and the BrdU labeled cells in the hippocampal region of the vmPFC stimulated middle-aged animal (see Figure 5L). We are convinced that this would provide strong evidence to show the effects of vmPFC HFS on the hippocampal neurogenesis-induced memory enhancement. For description of results, see the end of the subsection “Stimulation-induced changes from cell proliferation to dendritic spines modification” and Figure 6’s legend. For immunofluoresence staining methods, see Materials and methods (subsection headed “Immunofluorescence staining”). We hope that the data provided from the immunofluorescence experiments are convincible to the reviewer.

Reviewer 2:

*The description of the NOR study did not indicate whether the position of the new object and the familiar object are counterbalanced (so that the effect cannot be due to the animals preferring a certain location independently of the type of object placed in this location). If this was the case, the authors should describe this in the Material and methods section. If not, it should be discussed in the Discussion why this could not be the case and/or evidence for lack of spatial bias of the animals for the location where the new objects were placed should be provided (e.g. during the habituation, exploration time spent without object in the different quadrants should be measured and the time spent in the quadrant where will be placed the new objects shown not to be significantly different from that spent in other quadrants)*.

The reviewer commented the incomplete description of the novel object recognition task. During the test, rats were placed in the open-field with one familiar and one novel object (counter-balanced across tests and subjects) and given 3min to explore the two items. This description is covered in Materials and methods under the subsection headed “Behavioral Testing”.

Reviewer 3:

*1) There were a few areas where I thought the paper could be made clearer. First, the use of English needs to be improved. Second, more information is needed about how the stimulation was administered. For the acute stimulation study, it appears that stimulation was given for 30min before testing; does that mean the 30 minutes immediately before the habituation phase, or before each of the test phases? If it were the latter, stimulation would presumably affect retrieval processes and not encoding. Also, in the chronic stimulation study, it would be useful to know when the 1h of stimulation was given—at a set time of day? When did the stimulation occur relative to behavioral testing? I also would like to see explained what the no-HFS and HFS conditions referred to- perhaps, this condition refers to whether the stimulation session occurred just before the behavioral testing? Given this was a behaviorally relevant variable it should be spelled out clearly. In short, I do not think the stimulation procedure was described in sufficient detail for replication. Perhaps a figure given the sequence of events of all behavioral procedures would be useful*.

In response to the two concerns highlighted by the reviewer, we have taken the following measures. First, the manuscript has been thoroughly revised and proofread by a fluent English speaker. In response to the second concern about the insufficient information of the behavioral experimental studies, we have addressed the related issues in the Materials and methods section. (For information of acute and chronic stimulation prior to testing and the stimulation time, see subsection “Behavioral Testing” and for the explanation of no-HFS and HFS conditions, see the third paragraph of the same subsection). Finally, a figure of event sequence for all behavioral procedures has been inserted into Figure 3.

*2) I felt that the presentation of the correlation data was a bit muddled. I did not see a lot of value in computing an entire correlation matrix on this dataset with a relatively low N. A more useful approach might be to simply explore hypothesis-driven correlations (e.g. neurogenesis measures with behavior). I also do not think it is useful to present correlations with both sham and stimulated animals. If the hypothesis is that the benefit of stimulation is related to the degree to which it enhances neurogenesis, there is no reason to include sham animals*.

The reviewer commented that the data presentation of all correlation matrixes for both the sham and stimulated animals were seemingly not useful. Agreeing to this suggestion, we have removed the correlation on Table 1 and explored only the hypothesis-driven correlations of neurogenesis measures with behavior.

*3) Finally, I had two conceptual issues. First, the study is strongly motivated as addressing memory impairments as a result of age-related dementia. However, the animal model used here (12 month old rats) would seem to be a better model of healthy age-related memory decline. In fact, it seems the hypothesized mechanisms may not be effective when significant neural degeneration is present. Second, it may be worth discussing the relationship between mood effects of chronic ventromedial prefrontal cortex stimulation and the memory effect reported here. It may be that the stimulation has an effect on reducing anxiety, which increases novelty seeking and reduces thigmotaxic behavior in the Morris water maze, both which would enhance performance. The authors may want to consider whether the effect of ventromedial prefrontal stimulation on memory could be mediated by its anxiolytic action*.

In the first conceptual issue, we agree with the reviewer that the animal model used in this study would be an appropriate model of healthy aged-related memory decline instead of neural degeneration as described in our hypothesized mechanisms. We have revised and adapted the statements in the manuscript accordingly.

As for the second conceptual issue, the reviewer has suggested that the memory enhancement effects of ventromedial prefrontal cortex stimulation could be mediated by its anxiolytic action. In agreement with this suggestion, we have included these comments in the Discussion part.

[Editors' note: further revisions were requested prior to acceptance, as described below.]

*[…] With respect to the authors reply to Reviewer 1, since the authors were unwilling to test whether blocking the DBS-induced enhancement in neurogenesis would prevent the memory enhancement, I would suggest they add a 1-2 sentence discussion noting that they have not excluded that these effects are mediated by non-neurogenic mechanisms and identify one or two possibilities*.

With respect to your request to add 1-2 sentence(s) concerning the non-neurogenic mechanism, we have now included this within the Discussion section.